

# Sperm quality of artificially matured shortfinned eel is not affected by human chorionic gonadotropin dose and route of administration

Sean L. Divers, Sheri L. Johnson and P. Mark Lokman

Department of Zoology, University of Otago, Dunedin, Aotearoa New Zealand

## ABSTRACT

**Background.** Acquisition of high quality sperm is key to the artificial propagation of eels in captivity, but fertility drugs are expensive and repeated handling is stressful to the fish. An interrupted treatment regime (an initial hormone injection to stimulate spermatogenesis, followed several weeks later by weekly booster injections to induce sperm maturation) for acquisition of sperm in captive male eels has promise for high sperm quality on the one hand, and animal welfare benefits on the other. To further develop this approach for shortfinned eel, *Anguilla australis*, we evaluated the efficacy of (i) different initial doses of human chorionic gonadotropin (hCG) and (ii) route of administration.

**Methods.** Male eels were artificially induced to mature with a single injection of 0, 250, 500 or 1,000 IU/fish of hCG, administered either intramuscularly (IM) or intraperitoneally (IP). Sperm maturation was induced with 150 IU hCG/fish from week 5 onwards and sperm collected for evaluation of quality by computer-assisted sperm analysis.

**Results.** Control males did not mature and hence, sperm could not be retrieved and analysed, but all other treatments were effective in inducing testicular maturation. Milt volume tended to be higher for fish injected IM compared to those injected IP, whereas hCG dose had no effect. Conversely, the concentration of spermatozoa tended to be higher for several sperm collection time points in IP-injected than in IM-injected fish. Sperm quality, represented by percent motility, percent progressive motility and curvilinear velocity, was equal in fish given an initial dose of 250 IU hCG to those given higher initial doses of hCG.

**Conclusions.** We recommend that an initial dose of 250 IU hCG/fish be administered to induce spermatogenesis in male *A. australis*, and, after a period of 4–5 weeks, weekly booster injections of ∼150 IU hCG/fish be administered in the day prior to sperm collection; both routes of administration (IM or IP) are equally effective. We contend that an interrupted treatment regime has notable benefits for induced maturation in male anguillids, as it reduces fish handling and manipulation and reduces the resources required to produce high quality sperm.

Corresponding author
P. Mark Lokman,
mark.lokman@otago.ac.nz

# INTRODUCTION

Freshwater eel (*Anguilla* spp.) populations are in decline throughout the world, particularly in the Northern Hemisphere (*Tzeng, 2016*). With a ready market for eel-based food products throughout Europe and Asia, a significant amount of recent research has focused on closing the freshwater eel lifecycle in captivity to replace wild harvests. Most of this research effort has focused on the complexities of oogenesis in order to tease apart the factors influencing egg quality. Nonetheless, sub-optimal performance of eel spermatozoa remains an important issue (*e.g.*, *Herranz-Jusdado et al., 2019a*), although low sperm motility (the traditional assessment of sperm 'quality') can still result in successful fertilisation. It is clear however, that an improvement in sperm quality will benefit artificial propagation endeavours.

Although it is possible for male eels to mature spontaneously in captivity, as was shown for the Japanese eel, *A. japonica* (*Matsubara et al., 2008*), this is the rare exception rather than the rule. Hence, the use of exogenous hormone preparations aimed at inducing sexual maturity is well established for the freshwater eel. Various gonadotropin preparations have been used successfully, but human chorionic gonadotropin (hCG) has been shown as among the most cost-effective, readily available and consistent ones to induce spermatogenesis (*e.g.*, *Boëtius & Boëtius, 1967*; *Gallego et al., 2012*; *Kagawa et al., 2009*). This hormone has been used extensively by many research groups to investigate fertility in the male eel (chiefly, *A. anguilla* and *A. japonica*) using both *in vivo* and *in vitro* approaches. Injection is the most widely used method of its administration *in vivo*, but osmotic pumps and cholesterol pellets have also proven effective in delivering hCG, at least to male Japanese eels (*Kagawa et al., 2009*). To obtain viable spermatozoa *in vivo*, the optimum time between hCG injections is approximately seven days (*Ohta & Tanaka, 1997*). For a general protocol, males are injected once per week, spermiation (sperm hydration, migration and maturation) typically first occurring after four to six injections. Milt volume and quality often peak around 9–12 weeks of hCG treatment and then stabilise for a few weeks when most experiments end (*e.g.*, *Ohta et al., 1996*; *Pérez et al., 2000*), or death occurs. When stripping males for their milt, it has been shown that the best quality sperm is retrieved around 24 h post-injection, at least in the case of *A. anguilla* and *A. japonica* (*Pérez et al., 2000*; *Ohta et al., 2001*).

Notwithstanding the plethora of *in vivo* maturation studies that have been done on male eels, we recently highlighted the lack of baseline data on the relative efficacies of various doses and of routes of administration during routine hCG injection (*Lokman et al., 2016*). We demonstrated that by four weeks after a single dose of hCG, shortfinned eels, *A. australis*, injected IM developed significantly larger testes than did IP-injected fish (*Lokman et al., 2016*).

Although spermiation can be easily induced in male anguillids (*e.g.*, *Asturiano et al., 2005*), large differences are still observed in individual responsiveness to induction and in quality of sperm produced. As noted by *Ohta et al. (1997)*, these differences are particularly pronounced when a single or a low total number of injections is used, implying that constant weekly injections result in a more consistent spermatogenic response.

More recent methodologies and technology, such as computer-assisted sperm analysis (CASA), have enabled finer details about spermatozoa traits (*e.g.*, progressive motility, swimming path velocity) to be correlated to various treatment regimens such as temperature, type of fertility drug treatment, etc. To the best of our knowledge, the only CASA information available regarding interrupted (*i.e.*, non-standard weekly injection) treatment of eels with hCG is that from *Asturiano et al. (2005)* on *A. anguilla*. These authors compared fish injected weekly with 1.5 IU/g BW with males injected every two weeks, with males injected until spermiation, with males injected only three times, and with males injected weekly at half this dose. There were reported effects on sperm volume and spermiation period, but interestingly, not on sperm motility parameters (quality). It therefore appears that although standard weekly injections have a higher success rate of inducing spermatogenesis and spermiation, it is still possible to produce the same quality of sperm using an interrupted method in European eel. However, it is not known how applicable these observations are to other eel species.

In our study, we set out to address two aims: (i) given the well-documented effects of handling stress on reproductive function and gamete quality in a broad range of phylogenetically distant fishes (*e.g.*, rainbow trout, *Oncorhynchus mykiss* (Salmoniformes), *Campbell, Pottinger & Sumpter, 1992*; silver perch *Leiopotherapon plumbeus* (Perciformes), *Denusta et al., 2014*), and because presumed negative effects of handling on induced spawning were documented for an *Anguilla* relative, *Conger myriaster* (Anguilliformes; *Fueda et al., 2019*), we sought to further develop the interrupted method for inducing maturation in shortfinned male eels by determining the effect of hCG dose on sperm quality, assessed by CASA. To do so, repeated manipulation and costs of human resources were reduced by adopting a dose–response approach based on a single injection initially, followed a month later by weekly booster injections (c.f., *Koumpiadis et al., 2021*) to optimise sperm quality; and (ii) given that an effect of injection route (IM vs IP injections) has recently been shown on testis size (*Lokman et al., 2016*), we sought to evaluate whether this effect applied to sperm quality as well.

## MATERIALS & METHODS

### Animals

Forty male silver eels were captured from Lake Ellesmere, New Zealand, in autumn by commercial fishermen using fyke nets. The eels were packed into chilled polystyrene boxes and transported to Dunedin, where they were transferred directly into a 1,000 L recirculating tank of 35 ppt salt water, in keeping with the euryhaline abilities of these fish (*Damsteegt, Wylie & Setiawan, 2018*). Water temperature was gradually increased from 10 to 20 °C during a one-week period. The fish were held under a 12:12 (L:D) photoperiod with the tank covered in shade cloth to decrease light penetration and were monitored for activity and wellbeing at least once daily throughout the acclimation and experimental periods. PVC pipes were provided for shelter and refuge. Average body weight of the males at Day 0 was 138.6 g (± SE 2.6 g), with a range of 120–184 g. Fish were not fed during the experiment. The fish and manipulations used in this study were approved by the

University of Otago Animal Ethics Committee (AEC 36-11) in accordance with national welfare regulations.

## Experimental design and induction of spermatogenesis

The 40 fish, each separate experimental units, were assigned randomly by the same researcher (SLD) to one of four hCG dose groups. After an acclimation period of two weeks, the fish were anaesthetised in 0.15 g/L benzocaine (Sigma-Aldrich, St Louis, MO) before receiving their initial hCG injection (Day 0) of 0 (control), 250 (low dose), 500 (medium dose), or 1,000 IU/fish (high dose). Within each dose group, the fish were randomly allocated to an injection administration route of either IM (dorsal muscle) or IP (ventral midline, halfway between pectoral fins and anus), so that each *dose × administration route* contained five fish (c.f., *Lokman et al., 2016*) of comparable body weights (group weights 142–168 g; $P > 0.13$).

To allow for tracking of individuals, a PIT-tag was implanted into the muscle just anterior of the dorsal fin at the time of the first injection. At that time, control fish received an initial injection of 250 μL of ice-cold saline, whereas low-dose fish were injected with 250 μL of ice-cold purified hCG (Pregnyl, Batch # 128300; Merck, Sharp and Dohme Corp., Auckland, New Zealand) at 1 IU/μL, medium-dose fish at 2 IU/μL, and high-dose fish at 4 IU/μL. The initial injections were administered according to the randomly designated route, *i.e*, either IM or IP. Following hCG administration, all fish were placed into a second 1,000 L tank, that was part of the same recirculation system; accordingly, on treatment days, animals were randomly netted from the tank, injected, and then moved to the 'other' tank on the same system, thus switching back and forth between both 1,000 L tanks on injection days.

## Induction of sperm maturation

Every seven days from Day 31 onwards (six treatment days in total, designated Weeks 5–10), all eels received a booster hCG injection of 150 IU/fish in 250 μL of ice-cold saline, administered according to the specific route allocated to the individual. Injections were made after inducing anaesthesia in benzocaine. The purpose of this booster was to maximise milt production and quality as described previously. Weekly booster injections continued until Week 10, when, as expected, controls started spermiating, signalling the end of the experiment and rendering comparisons of sperm parameters across initial hCG doses no longer appropriate—indeed, spermiation in controls was assumed to not be due to the key focus of the experiment (injection of hCG at Time 0), but rather, on subsequent spermiation-induction from Week 5 onwards. Once spermiation started in the control group, eels were therefore placed in overdose benzocaine (0.3 g/L) and euthanised by spinal transection.

## Milt collection

Milt was collected from males one day after administration of the hCG booster while under anaesthesia in 0.15% benzocaine. The posterior portion of the abdomen and the vent were wiped gently with a paper towel before rinsing the same area with distilled water. The same area was again dried lightly with paper towel before gentle abdominal pressure was applied

to express milt. A modified 15 mL plastic tube and three mL Pasteur pipette coupled to a venturi was used to collect expressed milt via suction. The collected milt was then decanted into pre-weighed microcentrifuge tubes for assessment of total milt weight (c.f., *Koumpiadis et al., 2021*), as weight of a small sample is much more easily and accurately determined than the corresponding volume.

Milt volume was extrapolated from milt weight (assumed 1:1 volume/weight) and expressed as a proportion of body weight to account for variation in fish size where:

Milt volume % BW $=$ (milt weight (g)/body weight (g)) $*$ 100.

Milt was stored on ice until assessment for sperm quality by CASA which was done as soon as practicable on the same day, within 2 h of collection. Limited by equipment availability, not all time points were analysed for motility parameters (Weeks 6 and 9 were not done).

## CASA

Previous research (*Gallego et al., 2013*) has established that sperm quality does not change between ejaculate portions within a collection (*e.g.,* the first portion of an ejaculate is of no better quality than the last portion of milt expressed by hand-stripping). Therefore, CASA analysis for this experiment was conducted on a sample from the whole milt collected.

Sperm activity parameters were quantified using milt diluted in fresh sea water (sea water not previously used for fish husbandry). Pilot trials established an optimal dilution and suspension ratio of 1 µL milt in 199 µL of sea water, similar to that reported by *Gallego et al. (2013)*. The milt was ice cold and the seawater was at culture temperature of 20 °C. The suspension was quickly mixed and 4 µL loaded onto a 20 µm Leja slide (Leja Products BV, Nieuw-Vennep, Netherlands). A stopwatch was started at the time of milt addition to the water (sperm activation), the slide being loaded as soon as possible thereafter, and, comparable to other studies on anguillid eels (*Butts et al., 2020*; *Koumpiadis et al., 2021*), the first measurement of sperm activity being taken at 30 s post-activation. Pilot trials (data not shown) established that there was no difference in sperm activity measured at 15, 20 and 30 s post-activation, 30 s being chosen as a practical time point for repeatability. Video footage was recorded from 20 s post-activation onwards, using a camera (XC-ST50, Sony, Tokyo, Japan) mounted upon an external phase-contrast microscope at 10× magnification (CX41 Olympus, Melville, NY, USA). Each sample was analysed in duplicate (dual chamber slides) using a computerised sperm analyser (CEROS, Hamilton Thorne, USA). On average, 203 spermatozoa were tracked for each sample (range: 106–255) for 0.5 s at 60 frames/s. Alongside sperm concentration (CONC), sperm activity measures included percent motility (MOT), percent progressive motility (PROG), straight line velocity (VSL), curvilinear velocity (VCL), average path velocity (VAP), while path character was determined by path straightness (STR) and linearity. The lower threshold to determine static cells was predetermined at 20 µm/s for VAP and 17 µm/s for VSL. Progressive cells were pre-determined to have a minimum threshold VAP of 50 µm/s and 50% STR. Pilot trials established VCL as the best potential indicator of eel sperm activity, due to the spermatozoa swimming a slightly curved trajectory.
## Statistical analysis

Only data from responding eels were analysed, *i.e.*, non-responders (no sperm collected) were omitted *a priori* from analysis. Accordingly, animals in the control groups were not used in any of the analyses from Weeks 5–9, as sperm could not be obtained from these fish. Sperm that was collected from controls in Week 10 was deemed to have been produced not in response to treatment with saline, but in response to the weekly hCG booster injections starting in Week 5. These controls were, therefore, also excluded from statistical analysis, but data from controls were graphed for contrast against responders in hCG-injected groups.

Prior to analysis, all data were tested for normal distribution and homogeneity of variance (Levene's test), and subjected to log-transformation as required. For all motility data, angular transformations (arcsin $\sqrt{x}$; *Claringbold, Biggers & Emmens, 1953*) were carried out, as percent data are not normally distributed and two-way non-parametric tests are often problematic (*e.g., Mundry & Fisher, 1998*). Using IBM SPSS Statistics v.24, data were subsequently analysed using a series of full factorial two-way univariate ANOVAs, with route of administration (two levels: IM, IP) and hCG dose (three levels: 250, 500, 1000 IU hCG/fish) as fixed factors; Tukey's multiple comparisons were used to identify which contrasts differed when significant main effects were detected. The differences between groups for the dependant variables were analysed within week only, not between weeks. Data from Week five were excluded from statistical analysis, as only few eels responded in each treatment group; data from these fish have been included for graphical representation only. To determine whether there were differences in the number of fish that responded to each treatment, a Fisher's Exact Test was employed. Differences between means for all ANOVAs were considered to be significant for $P < 0.05$. In order to correct for Type I errors associated with running multiple ANOVAs (for each of three time points) for each sperm quality trait, the Benjamini–Hochberg's false discovery rate (FDR) was set at 5%.

## RESULTS

No mortality occurred, all fish appearing healthy throughout the experimental period.

### Number of responders to treatment with hCG

There was no difference in the number of fish that responded to the various treatments (Fisher's Exact Test, $P$ range = 0.42–0.96). Aside from control fish, every eel spermiated at least once during the trial. Only two fish did not spermiate every week once they had started; one fish in the IM medium-dose group spermiated in Week 5, but then only once more after that, in Week 9. A second fish, belonging to the IM high-dose group, first spermiated in Week 7, but then skipped a week before continuing to spermiate at the Week 9 and 10 time points. The first control fish to spermiate was encountered in Week 10, one day after the 6th booster injection of 150 IU (Fig. 1A).

### Milt volume

No significant main or interaction effects were seen when comparing relative milt volumes (Fig. 1B). Only during Week 10 was a strong trend ($P_{\text{route}} = 0.059$) detected, relative milt

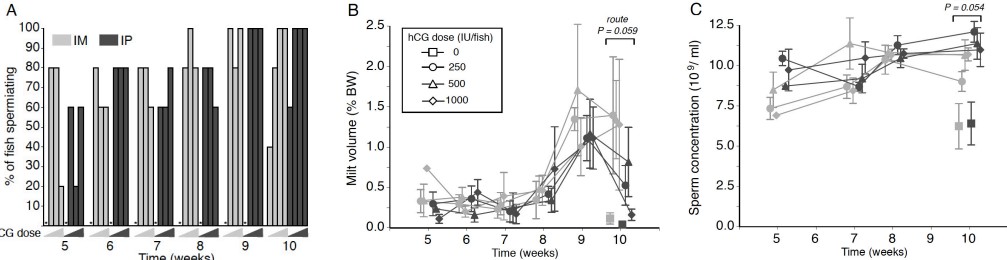

**Figure 1  Effects of human chorionic gonadotropin injection on sperm quantity parameters in eels (*Anguilla australis*).** Eels received an initial single injection of hCG (0, 250, 500 or 1,000 IU/fish), followed by a weekly booster (150 IU/fish) of hCG beginning at Week 5. Milt samples were taken 24 h post-booster. (A) The percentage of eels that responded to intramuscular (IM), or intraperitoneal (IP) injection with human chorionic gonadotropin (hCG), as determined by recoverable milt. Group size: $n = 5$. Increasing hCG doses from 0–1,000 IU/fish are represented by triangles of increasing height. Asterisks denote absence of any responders. (B) Milt volume ($\pm$SE) relative to body weight (BW) and (C) sperm concentration ($10^9$ spermatozoa ml$^{-1}$ $\pm$ SE) in milt samples in eels that responded IM (grey symbols), or IP (black symbols) injection with hCG. Only data from spermiating eels ($n = 1$–5) are presented, non-responders (no milt) being excluded from analysis. Note that symbols for each week are presented slightly off-centre to allow for easier tracking of individual treatment groups.

volume (roughly 1.5% of body weight) in IM-injected eels being almost double that of IP-injected eels; injection dose, on the other hand, did not have any effect ($P_{dose} = 0.256$). During Weeks 9 and 10, particularly large variation was observed in the amount of milt expressed between individual fish within each time point (Fig. 1B), which more than likely accounted for the lack of a significant treatment effect. The largest volume of milt recovered was 4.45% of body weight from one fish during Week 10 (high dose, IM), and from another fish (4.06%; medium dose, IM) in Week 9, which was in both cases several-fold greater than the average milt volume for IM-treated eels. This latter fish also produced a milt sample of 3.11% during Week 10, more than 2.5 times the average for IM-treated eels.

## Sperm concentration

Throughout the experiment, sperm concentration remained relatively stable at 8–12 $\times 10^9$ ml$^{-1}$ in all treatment groups, although a near-significant main effect of injection route was evident at Week 10 ($F_{1,23} = 4.136$, $P = 0.054$; Fig. 1C); thus, independent of dose, fish injected IP (Week 10: $11.48 \pm 0.50 \times 10^9$ ml$^{-1}$) produced sperm with a spermatozoa concentration around 15% higher than that observed in fish injected in the muscle (Week 10: $10.19 \pm 0.43 \times 10^9$ ml$^{-1}$). HCG dose and route of injection did not interact at any of the time points.

## Percentage of motile sperm

Weak treatment effects were seen when looking at percentage of motile sperm in a sample, an initial dose of 250 IU hCG appearing to be the most effective in terms of sperm motility throughout the experiment, particularly when administered IP (Fig. 2A)—this is especially evident from the near-significant effect of dose at Week 7 ($F_{2,15} = 3.223$, $P = 0.068$), when low-dose IP fish showed a relative motility twice that of fish in the remaining treatment

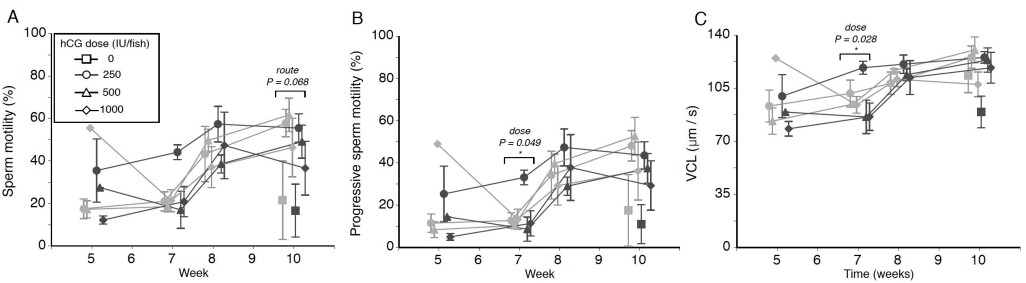

**Figure 2 Effects of human chorionic gonadotropin injection on sperm quality parameters in eels (*Anguilla australis*).** Eels received an initial single injection of hCG (0, 250, 500 or 1,000 IU/fish), followed by a weekly booster (150 IU/fish) of hCG beginning at Week 5. Milt samples were taken 24 h post-booster. (A) Mean percentage (±SE) of motile sperm, (B) Mean percentage (±SE) of progressive sperm motility, and (C) Mean curvilinear velocity (±SE). Only data from spermiating eels ($n = 1$–5/group) are presented, non-responders (no milt) being excluded from analysis. Note that symbols for each week are presented slightly off-centre to allow for easier tracking of individual treatment groups. Significant differences within each week are denoted by asterisk, and designated as either a route or dose effect. See text for implications of multiple statistical testing.

groups (44.1 ± 3.5% as opposed to 17.0–26.7% in other groups). There was no evidence for any significant interaction between hCG dose and route of administration (all $P > 0.05$).

## Progressive motility

The progressive motility data trends closely mirrored those seen for total motility percentage, low-dose IP fish displaying a generally higher level of progressively motile sperm throughout the experiment (Fig. 2B). Only during Week 7, a significant dose effect ($F_{2,15} = 3.717$, $P = 0.049$) was seen, the fish injected with the smallest amount of hCG showing more progressively motile sperm (21.5 ± 4.8%) compared to eels in the medium-dose (9.6 ± 2.4%) and high-dose (12.2 ± 4.6%) groups. However, this effect was no longer significant when taking into account that multiple tests were conducted. Significant interaction effects, similarly, were not observed.

## VCL

Finally, when focusing specifically on sperm swimming speed represented as curvilinear velocity (Fig. 2C), the only significant effect detected ($F_{2,15} = 4.569$, $P = 0.028$) was seen during Week 7, when fish injected with the lowest dose of hCG showed a higher average VCL (108.9 ± 6.1 μm s$^{-1}$) than in response to the higher doses (medium dose 91.1 μm s$^{-1}$, high dose average 89.7 μm s$^{-1}$). When applying the Benjamini–Hochberg correction for multiple statistical testing, this effect was no longer significant.

## DISCUSSION

Artificial acquisition of good quality sperm from eel is essential to ensure that artificially matured eggs can be fertilised so as to complete the life cycle in captivity. We aimed to combine the perceived benefits of repeated hCG injections on sperm quality on the one hand, and the welfare benefits of reduced handling and manipulation on the other.

For that purpose, we evaluated the suitability of an interrupted method that induced spermatogenesis with a single hCG injection, and then stimulated spermiogenesis and spermiation on a weekly basis; this approach mimics the repeated use of males 'on demand', as may happen during a prolonged artificially induced spawning season. Accordingly, the effect of hCG dose and route of administration were studied as our key variables.

All eels, regardless of treatment group, spermiated at least once, indicating that all doses and both routes of injection are viable methods of inducing complete spermatogenesis. No mortality occurred and all the fish were deemed to be in very good condition. This is in contrast to some weekly-injection experiments in which high mortality (>30%) has been reported (*e.g.*, *Pérez et al., 2000*).

The volume of milt retrieved from eels in all treatment groups followed the pattern typically seen in other studies on maturation in male eel (*e.g.*, *Asturiano et al., 2005*; *Pérez et al., 2000*), with a specific increase in milt collection from Week 8 onwards. Increased sperm yields are generally maintained during subsequent weeks. When considering the raw weight of milt collected from each group during the experiment, recoverable milt volumes were equivalent to those obtained from the Japanese eel (*Ohta & Unuma, 2003*), but lower than what can be achieved with the European eel (*Gallego et al., 2012*) in the same time frame, albeit using different hCG treatment regimes. Unlike the study by *Kagawa et al. (2009)*, who reported on a clear dose response for GSI and milt volume to hCG administered by osmotic pumps (1–50 IU/day), no clear dose response was evident in the present study. However, there was a trend in Week 10 for IM-injected fish to produce more milt than those injected IP, regardless of dose, with an apparent decline in milt produced by all IP injected fish. Although no gonad weight data were collected, a larger mature gonad relative to body size is capable of producing a proportionally larger volume of milt (*Kagawa et al., 2009*). This reinforces the findings reported by *Lokman et al. (2016)* on significantly larger testis relative to body weight in eels IM-injected with a single dose of hCG compared to IP-injected eels.

A trend for larger milt volumes in IM-injected eels coincided with significant reductions in spermatozoon concentrations in Weeks 5 and 10 compared to IP-injected fish. It is tempting to speculate that the higher volume and lower concentration of spermatozoa is due to different degrees of sperm hydration, a process mediated by progestins (see review by *Mylonas, Duncan & Asturiano, 2016*). Further research is required to reveal how the injection route might influence sperm concentration. Regardless, the overall sperm concentrations measured in eels from all treatments throughout the experiment were equivalent to those reported by *Gallego et al. (2012)*, averaging around 10 billion sperm/mL. The best sperm to egg ratio for fertilisation success in the European eel was around 25,000:1 (*Butts et al., 2014*). For fertilisation trials with *A. australis*, the current protocol would need to be implemented at least nine weeks in advance of milt requirements in order to secure appropriate volumes of milt at the concentrations recorded.

As all eels were co-housed, water-borne signals from hCG-injected individuals could have conceivably affected fish in the control group. *Huertas et al. (2006)* reported previously that hCG-treated male *A. anguilla* physiologically responded to spermiating or ovulating conspecifics, but that effects on gonadosomatic index (GSI) were "relatively minor". In our

previous study (Fig. 1 in *Lokman et al., 2016*), the GSI of control fish (0.155 ± 0.02%) was comparable to values reported for wild eels (0.20%; *Lokman & Young, 1998*), suggesting that any effects, if present at all, were minor. This deduction is further supported in said previous study (*Lokman et al., 2016*) in which co-housed eels treated with 11-ketotestosterone, the downstream mediator of hCG, did not generate measurable quantities of this steroid in the holding water.

There is currently no published information that correlates any particular CASA parameter to fertilisation success in the eel, but percentage of motile sperm and sperm velocity are generally considered the primary predictors of fertility in male fish (*Rurangwa et al., 2004*; review by *Kowalski & Cejko, 2019*; *Gallego & Asturiano, 2019*). In Atlantic salmon, sperm velocity was determined to be the primary component in fertilisation success during competitive trials (*Gage et al., 2004*). Like *Herranz-Jusdado et al. (2019b)*, we focused on motility, VCL and VSL as key parameters to characterise eel sperm by CASA, because spermatozoa moved in a slightly curved trajectory upon activation; VAP, LIN and STR were, therefore, not deemed important indicators. All three key motility parameters indicated that good quality sperm was obtained, particularly from Week 8 onwards, values retrieved being equivalent to, or better than, those reported for the European eel, from which CASA data are available (*Asturiano et al., 2005*; *Gallego et al., 2012*; *Vílchez et al., 2016*; *Herranz-Jusdado et al., 2019a*).

When examining trends in the motility data throughout the experiment, the lowest hCG dose, regardless of route of administration, resulted in comparable performance across total and progressive motility, as well as VCL, to performance in the medium and high hCG dose groups. Our findings are at odds with those from *A. japonica*. In this species, a dose of 250 IU hCG/fish yielded a larger milt volume than 750 IU hCG/fish, but data on motility were not presented (*Ohta & Unuma, 2003*). In contrast, when artificially inducing maturation with osmotic pumps in the same species, no differences in sperm motility were apparent when comparing eels treated with doses between 5 and 50 IU/day (*Kagawa et al., 2009*). To the best of our knowledge, the only other study that evaluated an hCG dose effect on sperm motility is that by *Asturiano et al. (2005)*. Their study similarly did not identify any differences in sperm motility between European eels injected weekly with 0.75 IU/g BW and those injected weekly with 1.5 IU/g BW for three months.

## CONCLUSIONS

We sought to evaluate the suitability of treatment with a single dose of hCG (0, 250, 500, 1000 U/fish) to induce testicular maturation in eel, followed by weekly injections with hCG (150 U/fish) to induce sperm maturation by two different routes of administration: IP and IM. Injection of a low-medium initial dose of hCG yielded sperm of at least equal quality to that of a high dose. Accordingly, for males in the size range of 100–200 g BW, it is recommended that a single injection of 250 IU hCG/fish be administered to induce spermatogenesis. Subsequently, after a period of four–five weeks, weekly injections of around 150 IU hCG/fish are effective to ensure spermiation is accompanied by high sperm quality. The route of injection does not appear to affect sperm quality, but IM injections

are recommended to maximise milt volumes. For future fertilisation trials using freshly caught silver male eels, the best volume and quality of sperm can be obtained after eight or more weeks of hCG treatment.

## ACKNOWLEDGEMENTS

We gratefully acknowledge Professor Neil Gemmell (Department of Anatomy, University of Otago) for the use of CASA hard- and software.

### Funding

This study was made possible through a University of Otago Doctoral Scholarship (Sean L. Divers) and a University of Otago Research Enhancement Grant (P. Mark Lokman). The funders had no role in study design, data collection and analysis, decision to publish, or preparation of the manuscript.

### Grant Disclosures

The following grant information was disclosed by the authors:
University of Otago Doctoral Scholarship.
University of Otago Research Enhancement Grant.

### Competing Interests

Sheri L. Johnson is an Academic Editor for PeerJ.

### Author Contributions

- Sean L. Divers conceived and designed the experiments, performed the experiments, analyzed the data, prepared figures and/or tables, authored or reviewed drafts of the article, and approved the final draft.
- Sheri L. Johnson performed the experiments, authored or reviewed drafts of the article, and approved the final draft.
- P. Mark Lokman analyzed the data, prepared figures and/or tables, authored or reviewed drafts of the article, and approved the final draft.

### Animal Ethics

The following information was supplied relating to ethical approvals (*i.e.*, approving body and any reference numbers):
University of Otago Animal Ethics Committee.

### Data Availability

The raw data are available in the Supplemental File.

### Supplemental Information

Supplemental information for this article can be found online at http://dx.doi.org/10.7717/peerj.13742#supplemental-information.

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
