# Peer review of "Sperm quality of artificially matured shortfinned eel is not affected by human chorionic gonadotropin dose and route of administration"

_PeerJ, doi:10.7717/peerj.13742_

## Round 0.1 · original submission · Minor Revisions

Your manuscript needs some minor modifications, mainly in the materials and methods sections. Please, see the reviewers' comments for more details.

·

Basic reporting

No comment

Experimental design

No comment

Validity of the findings

No comment

Additional comments

The MS is well written and it is organized and clear. It has an intrinsic interest for a better understanding of the reproductive biology of the species and an apply interest for its application in reproduction trials, cryopreservation protocols, etc.
However, I have some comments and questions to the authors:
L133-152: I have a comment about the experimental design that I would like the authors to consider. In the MS, the authors explain that all fish from the different experimental groups (hCG initial dose and administration routes) were held together in the same tank (1000L tank). I wonder if the authors have considered the possible effect of induced maturation from those individuals that matured faster (for instance, those treated with high dose of hCG) to immature individuals (those with low doses). That kind of effect have been reported previously. See for instance: "Huertas, M., Scott, A. P., Hubbard, P. C., Canário, A. V., & Cerdà, J. (2006). Sexually mature European eels (Anguilla anguilla L.) stimulate gonadal development of neighbouring males: possible involvement of chemical communication. General and comparative endocrinology, 147(3), 304-313.
L180: When the authors state that the analysis with CASA was carried out the same day, how long was after sperm extraction? Is there big variation in this “waiting time” between the different sampling weeks? Do the authors assume there is no effect of storing time on the sperm quality?
L198-210: Which CASA software did the authors used?
L216-219: Although I agree with the assumptions of the authors that the controls produced sperm due to the weekly hCG injections, still there is an interest in comparing its quality with the experimental treatments (and show their improvements). I recommend to analyze the data from the controls compared to the treatments and present them.
L354: The discussion approaches the comparison of A. australis sperm quality parameters with other from A. anguilla and A. japonica, but I believe it would be interesting to present a comparison with a weekly treatment in A. australis, (no initial injection but weekly injections since week 1 as those referred in the other anguillid studies) testing whether the studied protocols are better (or equal or worse) in terms of sperm quality. Do the authors know if there are studies reporting this results in A. australis?
L364. There are other studies that tested the dose effect of hCG on sperm quality. For instance Herranz-Jusdado, J. G., Rozenfeld, C., Morini, M., Pérez, L., Asturiano, J. F., & Gallego, V. (2019). Recombinant vs purified mammal gonadotropins as maturation hormonal treatments of European eel males. Aquaculture, 501, 527-536.
Comment on figure 1 text. The authors mention that "asterisks denote absence of any responder" and "significant differences... by asterisks), but in the graph there are no asterisks, so there is no point to mention that.

Reviewer 2 ·

Basic reporting

No comment

Experimental design

No comment

Validity of the findings

No comment

Additional comments

The MS is providing new indications on the inducting protocol for eel reproduction. Given the highly endangered classification of all eel species, and the stressful and costly procedure required for artificial reproduction, getting to an effective, less handling requiring protocol is very important.
Additionally, the information given on sperm motility are new and important, in order to get to the higher, best quality larval production possible

·

Basic reporting

In general, Divers et al.´s manuscript is well written and contains useful scientific information.

Experimental design

The number of eel males/treatment is quite low, what can compromise the strength of the conclusions. Moreover, some interesting parameters, as the length of the spermiation period or the total volume produced (even distinguishing between sperm qualities), has not been determined. However, the manucript brings an innovative approach to eel maturation hormonal treatment.

Validity of the findings

Practical conclusions of the experiment deserve their publication.

Additional comments

Considering the obtained results as a whole, do you still consider this “initial injection+booster doses” better than the traditional 150 IU hCG/fish weekly injections?. You could consider to use this treatment as a control in following experiments.

(L272 and next). Higher sperm concentration should not be directly understood as a signal of better sperm. In fact, a reduction of sperm concentration has been described in treated eels, coinciding with the final maturation process (sperm hydration regulated by MIS) and with higher sperm motility parameters.

You could consider these recent publications through the introduction (i.e.: use of osmotic pumps; use of purified vs recombinant hCG) and discussion (i.e.: reached sperm volume and concentration values):

Asturiano, J.F. Improvements on the reproductive control of the European eel. In: Reproduction in Aquatic Animals. From Basic Biology to Aquaculture Technology. M. Yoshida and J.F. Asturiano (eds.). Springer Nature. ISBN 978-981-15-2290-1 (eBook). Chapter 14. Pages: 293-320. 2020.

Blanes-García, M.; García-Salinas, P.; Morini, M.; Pérez, L.; Asturiano, J.F.; Gallego, V. Using osmotic pumps to induce the production of gametes in male and female European eels. Animals 2022, 12, 387. https://doi.org/10.3390/ani12030387

L32 Try to avoid sentences as this “tended to be higher”
L35 % (twice) > percentage of
L82 and next. You could mention here the Asturiano et al. (2005)´s reference as well
L99 You should consider the sperm volume produced as well, and even the length of the spermiation period.
L122 Were the fish directly moved from freshwater to seawater?
L124 10 to 20 °C
L146 Is this a recombinant or a purified hCG?. Results can be quite different.
L160 Why the experiment finished when the control fish started to spermiate?. Again, spermiation length or total produced sperm volume could be interesting parameters to compare hormonal treatments.
L189 Define “fresh seawater”
L208 This seems a risky assumption. Which were the criteria to compare the different parameters?
L246 What do you mean with “at least one”?
L261 The lack of significant differences could be due (at least in part) to the low number of fish per treatment.
L309 Maybe is more correct to talk about group-synchronous spermatozoa maturation through the reproductive season
L330 Without GSI values this is a risky assumption
L354 Vílchez
L354 Herranz-Jusdado 2019a showed some bigger values
L390 Garzón/Peñaranda/Jiménez/Martínez/Tomás
L404 captivity
L409 Is “quantal” right?
L438 724.
L440 ...Asturiano JF, Gallego V, 2019a
L444...Herranz-Jusdado JG, Gallego V,...
L428 Vílchez
L463 .........56, 1497-1505
L512 Tomás/Espinós

---

## Round 0.2 · accepted · Accept

Dear Authors,
I am pleased to confirm that your paper has been accepted for publication in PeerJ.

Thank you for submitting your work to this journal.